# Polycystin-1 Interacting Protein-1 (CU062) Interacts with the Ectodomain of Polycystin-1 (PC1)

**DOI:** 10.3390/cells12172166

**Published:** 2023-08-29

**Authors:** Wendy A. Lea, Thomas Winklhofer, Lesya Zelenchuk, Madhulika Sharma, Jessica Rossol-Allison, Timothy A. Fields, Gail Reif, James P. Calvet, Jason L. Bakeberg, Darren P. Wallace, Christopher J. Ward

**Affiliations:** 1Department of Nephrology and Hypertension, The Jared Grantham Kidney Institute, University of Kansas Medical Center, Kansas City, 3901 Rainbow Blvd., Mail Stop 3018, KS 66160, USAdwallace@kumc.edu (D.P.W.); 2Promega Corporation, 2800 Woods Hollow Road, Madison, WI 53711, USA; 3Department of Pathology & Laboratory Medicine, University of Kansas Medical Center, 3901 Rainbow Blvd., Mail Stop 3062, Kansas City, KS 66160, USA

**Keywords:** polycystin-2, autosomal dominant polycystic kidney disease (ADPKD), collecting duct, proximal tubule, C21orf62, migrasomes, exosomes, primary cilia, ligand

## Abstract

The *PKD1* gene, encoding protein polycystin-1 (PC1), is responsible for 85% of cases of autosomal dominant polycystic kidney disease (ADPKD). PC1 has been shown to be present in urinary exosome−like vesicles (PKD−ELVs) and lowered in individuals with germline *PKD1* mutations. A label−free mass spectrometry comparison of urinary PKD−ELVs from normal individuals and those with *PKD1* mutations showed that several proteins were reduced to a degree that matched the decrease observed in PC1 levels. Some of these proteins, such as polycystin-2 (PC2), may be present in a higher-order multi-protein assembly with PC1—the polycystin complex (PCC). CU062 (Q9NYP8) is decreased in ADPKD PKD−ELVs and, thus, is a candidate PCC component. CU062 is a small glycoprotein with a signal peptide but no transmembrane domain and can oligomerize with itself and interact with PC1. We investigated the localization of CU062 together with PC1 and PC2 using immunofluorescence (IF). In nonconfluent cells, all three proteins were localized in close proximity to focal adhesions (FAs), retraction fibers (RFs), and RF-associated extracellular vesicles (migrasomes). In confluent cells, primary cilia had PC1/PC2/CU062 + extracellular vesicles adherent to their plasma membrane. In cells exposed to mitochondrion-decoupling agents, we detected the development of novel PC1/CU062 + ring-like structures that entrained swollen mitochondria. In contact-inhibited cells under mitochondrial stress, PC1, PC2, and CU062 were observed on large, apically budding extracellular vesicles, where the proteins formed a reticular network on the membrane. CU062 interacts with PC1 and may have a role in the identification of senescent mitochondria and their extrusion in extracellular vesicles.

## 1. Introduction

Autosomal dominant polycystic kidney disease (ADPKD) is the most common hereditary kidney disease, affecting approximately 1:800 individuals; it is mainly due to mutations in two genes: *PKD1* and *PKD2* [1,2,3]. The *PKD1* gene is responsible for 85% of cases and *PKD2* for the remaining 15% of cases [4]. *PKD1* encodes polycystin-1 (PC1), a transmembrane protein containing 4302 amino acids (AAs), cleaved by an autoproteolytic event at 3048 AAs in the G−protein−coupled receptor (GPCR) autoproteolysis−inducing (GAIN) domain [5]. This cleavage produces a very large N−terminal ectodomain (450 kDa) and a C−terminal region that spans the membrane 11 times (150 kDa). The last six transmembrane (TM) domains (VI−XI) have homology with the protein product of the *PKD2* gene, polycystin-2 (PC2), an ion channel subunit. The ectodomain consists of a mosaic of domains, including a leucine−rich repeat (LRR), a solitary PKD domain, a C−type lectin, an LDL−receptor class A (LDL−A) motif followed by an array of 16 PKD domains, and a receptor egg jelly (REJ) composite domain that contains the GAIN domain [6,7,8].

PKD domains (PFAM00801, CD00146, IPR000601) compose 34.7% and 49% of the PC1 open reading frame (ORF) and ectodomain, respectively, and were first described in PC1 [6]. They represent a novel non-immunoglobulin type of β−sandwich consisting of 80−90 AAs stabilized by hydrophobic interactions between two opposing sheets, which are themselves composed of three and four β strands. PKD domains are known to form strong high-affinity homotypic interactions but few heterotypic interactors are known [9,10].

Fibrocystin is the product of the *PKHD1* gene, which is responsible for autosomal recessive polycystic kidney disease (ARPKD). It is a large type-1 membrane protein consisting of 4074 AAs, with an ectodomain composed of 12 IPT/TIG domains (an immunoglobulin−like fold), one PA14 domain (homologous to the protective antigen 14 (anthrax toxin)), two G8 domains (recognized by eight conserved glycines), and two blocks of parallel β−helix (PbH) [11,12].

PC1 has been localized to the primary cilium, the focal adhesions (FAs), and the desmosome [13,14,15,16]. However, the hardest evidence, proteomics (mass spectrometry/mass spectrometry (MS/MS)), shows that PC1 is an extracellular vesicle protein. PC1, PC2, and fibrocystin have been shown by Western blot and MS/MS to be present in polycystic kidney disease exosome-like vesicles (PKD−ELVs) (100 nm diameter membranous vesicles secreted apically into the nephron lumen and urine) [17]. At a molecular level, the three proteins interact in a 2 MDa complex termed the polycystin complex (PCC) [18].

Focusing on PKD−ELVs, we reasoned that if PC1 was decreased in PKD−ELVs of individuals with *PKD1* mutations, other unidentified PCC proteins would also be decreased to a similar degree, and then could be tested for interactions with PC1. In our previous work, we measured the levels of 2008 PKD−ELV proteins in a cohort of 13 *PKD1* and 18 normal individuals using label-free MS/MS on highly purified PKD−ELVs. In individuals with *PKD1* mutations, nine (0.32%) proteins showed statistically significant reductions, including PC1 (which was reduced to 54% of the normal level) and PC2 (which was reduced to 53%) [19].

We delineated a small signal peptide-bearing protein CU062 (a product of the *C21orf62* gene, DUF4571), which was decreased to 0.44 of the normal level in PKD−ELVs from individuals with *PKD1* mutations. An overexpressed CU062 protein could immunoprecipitate (IP) endogenous PC1.

In this paper, we investigate the PC1/CU062 interaction, and define tissue distribution and subcellular localization of CU062 and other members of the PCC [13,15,16]. As ADPKD has been associated with mitochondrial dysfunction, we also examine the distribution of PC1 and CU062 in the presence and absence of carbonyl cyanide m−chlorophenyl hydrazone (CCCP), which is a mitochondrial proton gradient uncoupling agent [20,21,22,23]. Finally, we suggest a unification hypothesis for the disparate observations that PC1, PC2, and fibrocystin are present on extracellular vesicles and that mitochondria from cystic epithelial are biochemically and morphologically abnormal. We postulate that the PCC is intimately involved in sorting and transporting senescent mitochondria for extrusion in specialized extracellular vesicles termed ‘migrasomes’, a process called mitocytosis [24,25]. Defects in PCC-dependent mitocytosis are responsible for the accumulation of senescent mitochondria observed in ADPKD.

## 2. Materials and Methods

We used the polycystin-1 sequence available from https://pkdb.mayo.edu/welcome (accessed on 19 August 2023) the reference coding sequence, PKD1.txt. We utilized this sequence as it is compatible with the mutation database at the same site. The PKD domain architecture was derived from http://www.uniprot.org/uniprot/P98161 (accessed on 4 May 2016), which has 17 PKD domains contrasting the Bycroft study, where 16 PKD domains were predicted [10]. The difference was due to an extra PKD domain immediately before (N−terminal to) PKD repeat 2 in the Bycroft study. The Bycroft PKD array starts at **ALVLQ** (844 AAs); the UniProt array starts at **LSANA** (743 AAs). We commence our PKD constructs at **PAPGH** (731 AAs), predicted to be outside this initial divergent PKD domain [10]. This results in the Bycroft array of PKD repeats being shifted one place so that Bycroft PKD−II is now UniProt/Lea PKD−III; see Appendix A.

The sequence used for CU062 was sp|Q9NYP8|CU062|HUMAN.

### 2.1. Generation of Microhomology-Based Gene Disruptions with Cas12a

We disrupted *PKD1* and *PKD2* using a microhomology end joining-based strategy and Cas12a. We targeted renal cortical tubule epithelial (RCTE) and normal human proximal tubule cell line (NHPTK) cell lines using the protocols described in https://www.youtube.com/watch?v=WoJZEDFc56Y&t=8s (accessed on 19 August 2023) and https://www.youtube.com/watch?v=Um_f2xtE9o4&t=602s (accessed on 16 August 2023) [26,27].

### 2.2. Transient Transfection of Plasmid DNA

We used the 293t cell line and a polyethylenimine (PEI) technique described in https://www.youtube.com/watch?v=gG96sFe_Aq4&t=844s (accessed on 16 August 2023). When two proteins were investigated for interactions, their cDNA-encoding plasmids were first mixed, complexed with PEI, and then applied to the cells.

### 2.3. Total Cellular Membrane Preparation

Cells were washed twice, harvested by scraping in ice-cold PBS, and recovered by centrifugation at 4000× *g* for 10 min at 4 °C. Cell pellets were resuspended in a low ionic strength lysis buffer (10 mM Tris, 0.8 mM MgCl${}_{2}$, pH 7.5) with the cOmplete ${}^{®}$-EDTA proteinase inhibitor (04693132001, Roche) and subsequently frozen at −80 °C to increase the protein yield. Thawed cells were then homogenized on ice using approximately 60 strokes in a tight-fitting Dounce homogenizer. After removing nuclei (3000× *g*, 2 min), membrane proteins were pelleted down by centrifugation (18,000× *g*, 30 min) and were further stored in the IP (139 mM NaCl, 50 mM Tris, 1% IGEPAL) buffer with cOmplete${}^{®}$−EDTA free proteinase inhibitor.

### 2.4. Immunoprecipitation with V5, OLLAS, and FLAG Antibodies

The membrane protein was solubilized in the IP buffer and was incubated with epitope tag antibodies, either anti-FLAG M2 (F3165 Sigma), anti-SV5-Pk1 (MA1-34099 Thermo Fisher Scientific), or the anti-OLLAS (NBP1-06713SS NovusBio) antibody, overnight at 4 °C. The antibody was used at 1 μg/mL. To each overnight sample, 30 μL of protein A/G agarose beads (20,421 Pierce) in the IP buffer were added. Samples were rotated for another 2 h at 4 °C. The agarose beads were then washed three times with the IP buffer followed by protein elution in the LiDS sample buffer (NP0007 Thermo Fisher) either with TCEP (reducing conditions with TCEP) or without TCEP (non-reducing conditions without TCEP) (77720 Thermo Scientific).

### 2.5. Western Analysis

The anti-PC1 mAb used was anti-N−terminal PC1 7E12 (IgG1κ) [28]. The PC2 antibodies were the polycystin-2 antibody (YCE2): sc−47734 (IgG2aκ, AAs 687−754), polycystin-2 antibody (D−3): sc−28331 (IgG2bκ, AAs 689−968), and polycystin-2 antibody CJW_9D5 (IgG2bκ, AAs 242−468, MSSN−YVTT) produced to the TOP domain of PC2 between TM I and TM II for this paper; see Appendix A and Section 2.6.1.

The Western technique is explained in the following video: https://www.youtube.com/watch?v=A_T4eqDaN4A&t=2s (accessed on 16 August 2023). When we Western-blotted immunoprecipitates (IPs), we used probing antibodies that were directly conjugated to horse radish peroxidase (HRP) to avoid detecting the primary antibodies used in the initial IP. For example, V5::HRP (R961−25 Thermo Fisher), OLLAS::HRP (NBP1−06713H, Novus Biologicals), and FLAG::HRP (A8592, Sigma−Aldrich) are all directly labeled with HRP, avoiding the detection of primary antibodies.

### 2.6. Production of Monoclonal Antibodies

We produced codon-optimized (*E. coli* codon preference for highly expressed genes) sections of the cDNA-encoding human PC2 protein, AAs 242−468 (MSSN−YVVT) from TM I to TM II, and the entire ORF of CU062 AAs 20−219 (LNCF−TFIY), but without the signal peptide for the expression in *E.coli*. The proteins had N−terminal His${}_{6}$ tags for metal affinity purification. The proteins were purified by Co${}^{++}$ affinity chromatography (635652 TALON${}^{®}$ metal affinity resin) under denaturing conditions, then dialyzed into PBS. Ten female BALB/cJ mice (per antigen) were vaccinated and seroconversion-monitored by Western blotting on TagGFP2::PC2 (FP−191, pTagGFP2−C, Evrogen), or CU062::V5 (pCDNA−3) transiently expressed in 293t cells. The mice that had the highest IgG titers were selected for boosting, and hybridoma production was performed as described by de StGroth et al. [29]. We describe the protocols in more detail in https://www.youtube.com/watch?v=SIaUypLo_5I&t=1658s, https://www.youtube.com/watch?v=LykCFNnwd8E (accessed on 16 August 2023), and https://www.youtube.com/watch?v=dItzBb60eeM&t=3s (accessed on 16 August 2023). We obtained one clone, CJW_9D5 (IgG2bκ), for PC2, and two clones, CJW_2D11 (IgAκ) and CJW_6F3 (IgG1κ), for the CU062 protein; see Appendix A. We screened 1600 wells for the PC2 mAb and 2000 wells for the two CU062 mAbs. We had difficulty finding these good mAbs, perhaps because both proteins are highly glycosylated in the region of interest, and epitope shielding was an issue.

#### 2.6.1. Verification

In order to formally show that the PC1 mAb antibody, 7E12, and the PC2 mAb, CJW_9D5, were cognate to their respective antigens, we targeted two human renal epithelial cell lines, RCTE and NHPTK cell lines with Cas12a CRISPRs; see Appendix A and Section 2.1. Frame-shifting mutations were created in exons-3 and 7 of the *PKD2* gene; see Appendix A. The *PKD1* gene was a special case, where we deleted the entire gene using two Cas12a CRISPRs, one centromeric (5’) to the 50 kbp *PKD1* duplicon but telomeric of the *RAB26* gene, and the other in the 3’ UTR of *PKD1* (this deletes the entire *PKD1* gene− 60.6 kbp of chromosome 16); see Appendix A [2,26,27]. Western analysis of the *PKD1* and *PKD2* null clones showed that although the WT parent RCTE cells had PC1 and PC2 specific bands, the *PKD1* and *PKD2* mutant cell lines did not. In the case of the CJW_9D5 mAb, there was a non *PKD2* derived band at 85 kDa in the RCTE cells but this was not observed in NHPTK cells. Thus, CJW_9D5 is completely specific to the PC2 protein in the PT cells and PKD−ELVs from human urine; see Appendix A.

Neither RCTE nor NHPTK cells produced the detectable CU062 protein; thus, the Cas12a knockout strategy could not be used with mAb CJW_2D11 or CJW_6F3. However, both mAbs detected glycosylated and non-glycosylated proteins of the expected size on a Western blot of kidney tissue, and both detected the recombinant mammalian-expressed proteins very well; see Figure 3A,G.

### 2.7. Real-Time Reverse Transcriptase Polymerase Chain Reaction (RT−PCR)
for Mouse *C21orf62*

We generated a synthetic target cRNA for the *C21orf62* gene by cloning a 589 bp section of the mouse *C21orf62* ORF downstream of the T7 promoter in plasmid pT7. The target contained the genomic DNA from 90853107−90852519, derived from the Mus musculus strain C57BL/6J chromosome 16 in the GRCm39 genomic assembly. The plasmid was linearized and transcribed to RNA using a MEGAscript T7 Transcription Kit (AMB13345 Ambion), and then the DNA template was removed using a TURBO DNA-free Kit (AM1907 Ambion). Total RNA was measured using a Qubit 4 fluorometer and a Qubit RNA BR Assay Kit (Q10211 Thermo Fisher). A standard curve for copies per μL was generated from 0 copies μ$L^{-1}$–2 × 10 ^7^
copies μ$L^{-1}$.

#### RT−PCR

Total cellular RNA was prepared using TRIzol Reagent (15596026 Thermo Fisher) followed by DNA removal using a TURBO DNA-free Kit (see the above). Total RNA was measured using a Qubit 4 fluorometer. RT-PCR was accomplished using SuperScript IV Reverse Transcriptase (18090010 Thermo Fisher) on all samples, transcribing a maximum of 2.5 μg RNA per reaction and using the gene-specific primer TGTGGATGAGCAAACTCTGC at 100 fmol μL. We always performed an RT reaction with no reverse transcriptase to ensure that there was no contaminating genomic DNA. In all cases, the RT− sample was below the limit of detection for the real-time assay.

We developed a real-time TaqMan assay utilizing the forward and reverse primers, CAGCTTCTGCCTCCTTCTG and CTGCAGTTGCGAATGGTATTTC, respectively, and an internal FAM, labeled TaqMan probe 56−FAM/TTTGTGAAG/ZEN/TGGCCAAGGGCAAC/3IABkFQ/. The PCR program employed a start at 50 °C for 2 min, 95 °C for 10 min, followed by 40 cycles of 95 °C for 15 s, and 60 °C for 1 min on a 7500 real-time PCR system (Thermo Fisher Scientific) using PrimeTime Gene Expression Master Mix (1055772 IDT). The data analysis was performed using R https://www.r-project.org (accessed on 16 August 2023).

### 2.8. Immunofluorescence (IF)

Primary human renal epithelial cells, RCTE cells, or NHPTK cells, were plated on no. 1.5 cover slips coated with 10 μg m$L^{-1}$ fibronectin in PBS overnight, washed 1x in PBS, then air dried. The medium used was DMEM:F12 (1:1 #D5525 and #N6760 Sigma) with 15 mL per liter of HEPES (#H0887 Sigma) and 2 g $L^{-1}$ NaHCO${}_{3}$ with 5% fetal bovine serum (FBS) (#S11550 R&D systems), penicillin/streptomycin (P-0781 Sigma), insulin, transferrin, and selenium (#354351 Corning).

Cells were harvested at 50% confluency, when observing focal adhesions (FAs)/retraction fibers (RFs) and migrasomes. To observe decoupled mitochondria, we treated 50% confluent cells with 2 μM CCCP for 1–9 h. To observe primary cilia, we allowed the cells to become confluent and then serum-starved the monolayer in the above medium with 0.05% FBS for 2 days.

Fixation was with warm (37 °C) 4% paraformaldehyde and 0.15% glutaraldehyde in PBS for exactly 10 min, then quenched with 100 mM NH${}_{4}$Cl in PBS for 5 min, 2 times. Cells were then treated with 10 mg m$L^{-1}$ sodium borohydride (NaBH${}_{4}$ 452882 Sigma-Aldrich) in PBS for 30 min, which was then replaced for overnight incubation, then given another 30 min treatment to reduce the Schiff base autofluorescence. The cells were then permeabilized with 0.2% triton X−100 and 0.1% saponin (Quillaja bark S−4521 Sigma) in PBS for 15 min. We then used the protocol described in https://www.youtube.com/watch?v=iCb7S-sp2OQ&t=4517s (accessed on 16 August 2023).

#### Microscopy

4’,6−diamidino−2−phenylindole (DAPI) was used at 200 nM in PBS for 10 min to counter-stain nuclei. The mounting agent was ProLong${}^{®}$ Diamond Antifade Mountant (P36965 Invitrogen). The microscope used was a Leica SP8 confocal microscope with STED, and the objective was HC PL APO CS2 100X/1.4 oil. The acquisition was with the Leica Application Suite X (LAS X), 16 bits, with frame average 1, frame accumulation 3, in 2048 × 2048 pixel format.

### 2.9. Statistics

Tukey boxplots were performed using R, specifically ggplot2 https://www.r-project.org/ (accessed on 16 August 2023), using packages dplyr http://dplyr.tidyverse.org (accessed on 16 August 2023), https://github.com/tidyverse/dplyr (accessed on 16 August 2023), http://ggplot2.tidyverse.org (accessed on 160823), https://github.com/tidyverse/ggplot2 (accessed on 16 August 2023).

### 2.10. Cell Lines

The human renal cortical tubule epithelial (RCTE) cell line was a gift from Prof. Jing Zhou, Harvard Institutes of Medicine, Suite 520, 77 Avenue Louis Pasteur, Boston, MA 02115 [30]. The normal human proximal tubule kidney (NHPTK) cell line was a gift from Prof. Robert L. Bacallao Division of Nephrology, Richard L. Roudebush, VAMC, and Indiana University, Indianapolis, Indiana, United States of America [27]. Both cell lines were derived from anonymous donors.

## 3. Institutional Animal Care and Use Committee (IACUC) (Animal Work)

“KUMC is fully accredited by the Association for the Assessment of Accreditation of Laboratory Animal Care, International (AAALAC). KUMC is registered as a research facility with the United States Department of Agriculture (USDA) in accordance with the Animal Welfare Act and all amendments. KUMC also holds a Category I Assurance with the Public Health Service. Mouse vaccinations and monoclonal antibody work were covered under IACUC Protocol Registry number 2020–2551, PI: Dr. Christopher James Ward MBChB PhD”.

## 4. Results

### 4.1. The CU062 Protein Is the Product of the
*C21orf62* Gene

The *C21orf62* gene is on the q arm of Chromosome 21 (21q22.11; chr21 32790673– 32814181 (–) build GRCh38.p2) [31]. The gene is predicted to have four exons: the first three are non-coding, exons 2 and 3 can be skipped in different splice forms, and exon 4 contains the entire open reading frame (ORF); see Figure 1A. The CU062 protein is encoded by an open reading frame (ORF) of 219 amino acids, and is composed of a single, vertebrate-specific domain of an unknown function (DUF4571). It is predicted to have a 26-AA signal peptide that cleaves at KG↑QKN (↑ = cleavage site) but not a transmembrane domain; see Figure 1B [32]. The mature protein encompasses 193 AAs in humans, has a molecular mass of 22.1 kDa, an isoelectric point of 7.6, and five N-linked glycosylation sites, four of which are conserved. Glycosylation adds 18 kDa of peptide-N-glycosidase F (PNGase F)/endoglycosidase H (EndoH)-sensitive carbohydrates to the peptide backbone when the protein is expressed in 293t cells; Figure 1C. The CU062 protein is able to self-associate into oligomers, which are stable on non-reducing polyacrylamide gels (PAGE) in the presence of 0.5% lithium dodecyl sulfate (LiDS) and a temperature of 65 °C, Figure 1D.

### 4.2. The CU062 Protein Interacts with PKD Domains 2–17 from PC1

CU062 reciprocally IPs the PKD repeats of PC1 as dimers or trimers and demonstrates a high affinity interaction in the last three repeats (repeats 15–17); see Figure 2 and Appendix A. This interaction was robust and was resistant to 0.5% LiDS at 65 °C. Overlapping trimers derived from repeats 2–17 could all reciprocally IP CU062, Appendix A. The singleton PKD repeat, PKD−I, did not interact with CU062 and CU062 failed to IP single PKD domains. It requires at least a dimer PKD repeat for interaction; see Appendix A. CU062 could not IP other portions of PC1, including the region from the N−terminus to the beginning of PKD−4, the receptor egg jelly (REJ) domain, and the C−terminal membrane-spanning portion of the molecule; see Appendix A.

PKD repeats could homotypically interact under non-reducing conditions in the presence of 0.5% LiDS and a temperature of 65 °C, confirming the work of Ibraghimov-Beskrovnaya et al. in a mammalian system [9]; see Figure 2B and Appendix A.

#### 4.2.1. The PKD/PKD, PKD/CU062 and the CU062/CU062 Interactions Are Not Carbohydrate-Dependent

As the PKD repeats and CU062 are highly glycosylated, we investigated the possibility that the PKD/PKD, PKD/CU062 or CU062/CU062 interactions were ‘lectin-like’ and focused on domains PKD 2_5, PKD 8_11, and PKD 15_17 co-transfected with CU062. The cells were treated with the class 1 α-mannosidase inhibitor kifunensine at the time of transfection (to ensure susceptibility to EndoH deglycosylation) and EndoH prior to IP [33]. Deglycosylation caused both CU062 and PKD domain proteins to shift downward in molecular weight. In no case did deglycosylation interfere with the PKD/PKD, PKD/CU062, or CU062/CU062 interactions, implying carbohydrate independence; see Appendix A.

#### 4.2.2. The CU062 Protein Interacts Specifically with PC1 PKD Domains

To investigate whether CU062 could interact with PKD domains from other proteins, we utilized the observation that the dyslexia-associated protein, KIAA0319-like protein (KIAA0319L), has five PKD domains in the range of 312–785 AAs. We obtained a full-length cDNA with a C−terminal FLAG tag and showed that it could not IP or be IPed by CU062, implying a degree of interaction specificity between CU062 and PC1 PKD domains; see Appendix A.

### 4.3. Localization of the CU062 Protein

Given the controversy over the immunolocalization of PC1, we undertook immunofluorescence (IF) studies on both PC1 and CU062 proteins in human-derived tissue and primary cells.

Anti-CU062 mAb CJW_2D11 recognized a 42 kDa protein in human kidney membrane preparations as well as higher molecular weight bands that were likely to be non-glycosylated oligomers. In the kidney, the 42 kDa glycoform was sensitive to deglycosylation with EndoH and PNGase F, implying endoplasmic reticulum (ER) localization. However, the glycoform present in the PKD−ELVs was resistant to EndoH, implying post-Golgi carbohydrate modification; see Figure 3A. Further, CU062 co-resolved with PC1, PC2, fibrocystin, and CD133 (prominin) on urine-derived PKD−ELVs on a ‘float up’ heavy water (D${}_{2}$O), 5–30% sucrose gradient, implying that they are on the same 100 nm diameter PKD−ELV in vivo; see Figure 3B [17,34]. In confluent-polarized serum-starved human primary renal epithelial (HPRE) cells, CU062 colocalized with PC1 on the primary cilia. PC1 (7E12, IgG1κ) and PC2 (CJW_9D5, IgG2bκ) staining appeared as hollow 100 nm diameter vesicles attached to the primary cilium, and CU062 staining appeared as an outer shell around the PC1+ structure, implying close association; see Figure 3C,D and Appendix A. For antibody verification, see Section 2.6.1. In the collecting duct (CD) of the adult kidney, CU062 could be observed colocalizing with PC1 on the acetylated α−tubulin positive primary cilia; see Figure 3D,E. Again, PC1 and CU062 colocalized on 100 nm vesicles. It is likely that these vesicles are secreted by more proximal cells into the lumen of the nephron and interact with the primary cilia of CD cells [35]; see Section 4.4.2. CU062 was also observed in the endothelium of large blood vessels; see Figure 3F. Trace amounts of CU062 could be observed in circulating blood exosomes from a fasting individual; see Figure 3G. We assessed murine CU062 (*C21orf62*) mRNA levels in mouse tissues using an internal synthetic RNA standard to derive copies μ$g^{-1}$ of total cellular RNA. This showed that the transcript was rare in the liver (15,532 ± 7411) copies μ$g^{-1}$, three times higher in the kidney (45,499 ± 24,237) copies μ$g^{-1}$, and abundant in the testes (8,871,151 ± 1,746,007) copies μ$g^{-1}$; see Figure 3H.

### 4.4. Localization of PC1 and CU062 Depend on the Cell Type and Growth
Status (Confluent vs. Nonconfluent)

Apart from staining on the primary cilia, we did not observe PC1 or CU062 expression in normal human kidney tubules that were uniformly negative for the protein at this level of sensitivity (tissue section from a normal pediatric kidney); see Figure 3D,E. The CU062 protein could not be observed by Western blot or IF in our immortalized human kidney epithelial cells, human renal cortical tubule epithelial (RCTE) cells or normal human proximal tubule kidney (NHPTK) cell line cells; see Section 2.6.1. Human primary renal epithelial (HPRE) cells are freshly isolated primary cells from the human kidney, containing multiple cell types proximal tubule (PT), CD, the loop of Henle, (LOH), and endothelial cells, and both PC1 and CU062 are detectable by IF in these. The present study shows that the localization of PC1, fibrocystin, and CU062 depends on the tubular origin of the cells as well as their confluence status.

#### 4.4.1. Nonconfluent Migrating HPRE Cells

Focusing on PC1, we showed that PC1 completely colocalized with tetraspanin−4 (TSPAN4), a marker for retraction fibers (RFs) and migrasomes on a subpopulation of cells; see Figure 4A (top) [25,36,37]. These cells were megalin (*LRP2*) +, indicating that they are proximal tubule (PT) epithelial cells; see Figure 4A (bottom). Migrating PT cells left a trail of PC1+ retraction fibers across the fibronectin-coated substratum associated with TSPAN4/PC1/CU062 + migrasomes. Migrasomes are extracellular vesicles that are shed by migrating cells on RFs and are involved in the shedding of defective mitochondria [25,36,37]. CU062 appeared to form an external shell over a core of PC1 in these structures; see Figure 4C. PT cells were always PC1/CU062+.

CU062 was also found in proximity with talin and α-actinin-positive FAs, together with PC1 in nonconfluent PT cells. We also show that PC1+ FAs in PT cells come into close proximity with heat shock protein 60 (HSP60)+ mitochondria at the base of the cell; see Appendix A. PC2 and fibrocystin are also in proximity to the PC1 and CU062 staining with moderate colocalization, which is expected when the theoretical lengths of PC1 and fibrocystin are > 70 nm; see Figure 4B [15].

#### 4.4.2. Confluent HPRE Cells

Confluent and serum-starved HPRE cells develop primary cilia. We noticed that most acetylated α-tubulin-positive primary cilia had PC1+ extracellular vesicles attached to the shaft of the primary cilium. Considering their sizes at 100 nm in diameter and the finding that fibrocystin-positive PKD−ELVs can adhere to primary cilia, these are likely PKD−ELVs [35]. A subpopulation of the PC1+ primary cilia were associated with cells that did not have intracellular PC1; see Appendix A. Staining with *Lotus tetragonolobus agglutinin* (LTA) showed that all LTA+ cells were PC1+ and that most *Dolichos biflorus agglutinin* (DBA)+ cells were negative for intracellular PC1+. However, DBA+ CD cells often had PC1+ vesicles attached to their primary cilia despite having no intracellular PC1; see Appendix A. This implies that PT cells are the major producers of PC1+ extracellular vesicles and that CD cells may collect extruded PKD−ELVs/migrasomes on their primary cilia [35].

### 4.5. Localization of PC1 and CU062 with Mitochondria

Mitocytosis is the process where senescent mitochondria are identified, packaged, and removed from cells in 200 nm–500 nm TSPAN4 + extracellular vesicles, called ‘migrasomes’ [25]. Mitocytosis can be induced with low levels (2 μM) of CCCP, an H${}^{+}$ ionophore that abolishes the inner-membrane potential ($\Delta \psi_{m}$) of the mitochondrion. This phenomenon occurs at CCCP concentrations that do not induce mitophagy [25,36,37]. In ADPKD, physiological and ultrastructural studies have shown that there are multiple defects in mitochondrial functions and morphology [20,21,22,23,38]. It may be possible to unify these observations with the finding of the TSPAN4/PC1 colocalization on extracellular vesicles by suggesting that mitocytosis is defective in ADPKD and that low levels or the absence of PC1 or CU062 could be responsible for the accumulation of senescent mitochondria observed in cyst-lining cells; see Section 4.4.1.

We induced mitocytosis in HPRE cells with 2 μM CCCP for 1−9 h. In agreement with the work of Jiao et al., we observed PC1+ extracellular vesicles, migrasomes/PKD−ELVs, loaded with TOMM20+ material (TOMM20, a translocase subunit of the outer mitochondrial membrane) attached to RFs trailing migrating cells; see Appendix A. However, the major and unexpected feature of the experiment was the appearance of nidi within the cell that exhibited intense staining for PC1, PC2, and the CU062 protein; see Figure 5. Initial experiments where the cells were treated for two hours with 2 μM CCCP showed that PT cells developed nidi, followed by rings of PC1+ material with a central core of CU062. These rings were associated with swollen and disintegrating mitochondria but were too large to be autophagosomes and did not stain for autophagosome markers; see Appendix A. We refer to these 3μm–6μm PC1+ ring-like structures, as fairy rings (FRs). This is because the PC1+ ring resembles the exuberant grass and the mitochondria resemble the mushrooms. On the edge of the migrating cell, there are accumulations of PC1/CU062 and TOMM20 material, which are mitochondria being extruded in migrasomes; see Figure 5A,B. At four hours of CCCP treatment, and in confluent HPRE cells, we observed the release of large hollow extracellular vesicles 2μm–6μm in diameter. These had a wall composed of closely interacting reticular, PC1, PC2, and the CU062 protein, and never contained nuclear material and, thus, are not apoptotic in nature (nuclei were not fragmented); see Figure 5C–E (arrows). TEM of cells treated with CCCP showed apical vesicles with fragmented, damaged, and injured mitochondria, which contrast with the normal mitochondria seen in the rest of the cell; see Figure 5E. We believe that FRs are precursor structures involved in the detection and sorting of senescent mitochondria. Defective mitochondria are then extruded from cells in classical migrasomes in nonconfluent migrating cells and expelled apically in much larger vesicles in confluent epithelial cells, apical mitocytosis.

## 5. Discussion

This is an initial description of the CU062 protein and its interaction with PC1. CU062 is composed of a single domain of an unknown function (DUF4571), which is found only in vertebrates. It is conserved to *Chondrichthyes* (sharks, skates, and rays) and is not seen in other chordates (*Tunicata* and *Cephalochordata*).

CU062 was uncovered using label-free proteomics that compared the PKD−ELVs of normal individuals to those with *PKD1* mutations. Our hypothesis was that individuals with *PKD1* mutations would have less PCC, and this would be reflected in a decreased level of PC1 interactors, some of which could be novel. CU062 was decreased in ADPKD PKD−ELVs; it could interact with itself and PKD domains 2–17 in PC1, generating high order homo- and hetero-oligomers; see Figure 1, Figure 2 and Appendix A. The interactions between CU062 and PKD repeats of PC1 were robust, surviving treatment with 0.5% LiDS detergent and 65 °C for 10 min. The homotypic (PKD/PKD and CU062/CU062) and heterotypic (PKD/CU062) interactions were not lectin-like; they were specific to the PKD repeats in PC1 and not the repeats in the dyslexia-associated protein, the KIAA0319-like protein (KIAA0319L); see Appendix A.

The localization of PC1 has been controversial, with the protein being detected on the primary cilium, and in FAs, desmosomes and extracellular vesicles (ELVs) [13,15,16,17]. An important part of this study was to formally revisit the localization of PC1 while investigating the localization of the CU062 protein. Our efforts to localize these proteins showed that we could **not** detect PC1 or CU062 by IF in immortalized RCTE cells and had only weak staining for PC1 in NHPTK cells; see Section 2.10. Expression levels were much higher in freshly isolated primary HPRE cells. To consistently observe the sheddable PC1 and fibrocystin ectodomains, we found it necessary to use an efficient cross-linking fixative to retain the higher-order structures. Titration experiments showed that 4% paraformaldehyde and 0.15% glutaraldehyde in PBS was a good compromise between autofluorescence the and maintenance of protein−protein interactions in the PCC.

HPRE cells are non-immortalized primary cells derived from the kidney, consisting of a mixture of PT, CD, the loop of Henle, and endothelial cells. In migrating HPRE cells, PC1 and CU062 are expressed in the LTA/megalin (*LRP2*) + PT cells. In PT cells, PC1, PC2, fibrocystin, and CU062 colocalized with the FA proteins talin and α-actinin, as shown by Wilson et al. [15]. PC1 also colocalized with the RF and migrasome marker TSPAN4 and stained migrasomes intensely. The CU062 protein formed a shell of staining around the inner PC1 staining on the migrasomes.

In confluent cells, PT cells had intense intracellular staining (ER type) for both proteins; the cell bodies of DBA+ CD cells were negative for intracellular PC1/CU062, although the primary cilia of these cells had PC1/CU062 + vesicles attached to the primary cilia; see Appendix A. There is evidence that extracellular vesicles termed nodal vesicular parcels can interact with primary cilia in the transient embryological node and that fibrocystin + PKD−ELVs interact with the primary cilia of IMCD3 (CD) cells [35,39].

ADPKD is associated with mitochondrial dysfunction, and migrasomes have been shown to remove defective mitochondria in migrating cells upon challenges with low-dose CCCP [21,22,23,24,25]. When we challenged migrating HPRE PT cells with 2 μM CCCP, cells extruded mitochondria on PC1+ RFs in TSPAN4/PC1/CU062 + migrasomes; see Appendix A. However, the most salient observation was that PC1 was found at short time periods (two hours) in intensely staining nidi and at longer time periods in circular rings that were associated with swollen disintegrating mitochondria; the core of these structures was CU062+; see Appendix A. We denote these 3μm–6μm diameter structures as fairy rings (FRs) as they are circular and transient; the PC1 signal resembles the vibrant grass and the mitochondria resemble the mushrooms in these fungal phenomena. In Celtic mythology, the fairy ring is the interface between life and death, and the FRs are the source of the PKD−**ELVs**; thus, we believe the etymology is appropriate. We believe that FRs are composed of PC1/PC2/CU062 + ER interacting with senescent mitochondria and that they are involved in a sorting event that segregates mitochondria that have lost their inner membrane potential ($\Delta \psi_{m}$). These mitochondria are then extruded either as migrasomes on the RFs of the migrating cell or in the case of confluent, non-migrating cells in large 2 μm–6 μm apically secreted extracellular vesicles; see Figure 5C–E. A high-resolution analysis of these large vesicles shows that they are, in part, composed of a higher-order reticular assembly of PC1, PC2, and CU062 monomers in a hollow basket-like structure; see Figure 5C,D.

In summary, PC1 and CU062 are found:1.On the FAs of migrating PT cells.2.On the RFs, especially the migrasomes where CU062 forms a shell around the vesicle.3.In novel structures, FRs, induced by CCCP.4.As extracellular vesicles attached to the primary cilia of CD cells where they may have signaling roles.

Migrasomes have been implicated in the extrusion of defective mitochondria [24,25]. This process, termed mitocytosis, removes mitochondria that have lost their inner membrane potential ($\Delta \psi_{m}$) and are producing large amounts of reactive oxygen species (ROS). Poor mitochondrial functioning is one of the features of the cyst-lining epithelial cells in ADPKD, with a defect in the glucose entry into the tricarboxylic acid cycle (TCA) under aerobic conditions, the Warburg effect, a defect in β oxidation, and a dependence on the glutamine/glutamate/α-ketoglutaric/TCA pathway being important features [21,22,23]. Mitochondrial dysfunction may be due to a PC1-dependent defect in mitocytosis so that cyst-lining epithelia accumulate defective mitochondria with multiple abnormalities.

The mitochondrial phenotype is not the only cellular defect observed in ADPKD. The localization of PC1 and CU062 on the FAs and retraction fibers may also point to a global alteration in cytoskeletal functioning, as observed in the missorting of apical and basolateral proteins, the disruption of planar cell polarity (PCP), and the observation of polyploidy, centrosome amplification, and thickened basement laminae (BL) [40,41,42,43]. Thus, mitochondrial dysfunction is just one aspect of ADPKD, even though amelioration of the cystic phenotype is possible using mitochondrion-targeting drugs, such as 2-deoxyglucose [44]. CU062 may offer insight into a subset of functions of the PCC. For example, the CU062 function may be restricted to the sorting of senescent mitochondria and this function may be distinct from other cytoskeletal and BL remodeling and PCP roles [45].

## 6. Conclusions

Here, we show that CU062 interacts with PKD repeats 2–17 of PC1 and that it colocalizes with PC1 and PC2 on the primary cilia, FAs, and migrasomes in primary cells. Upon challenge with mitochondrial decoupling agents, such as CCCP, PC1 and CU062 both associate with swollen and fragmenting mitochondria in a higher-order structure, termed the FR. We believe that the PC1/CU062 complex is involved in the detection, sequestration, and ultimate exocytosis of senescent mitochondria. In migrating cells, the mitochondria are removed in classical migrasomes, and in confluent cells, they are extruded in large apical vesicles, the surfaces of which are in part composed of a macromolecular complex of PC1/PC2 and CU062. Future work will entail working with *PKD1*, *PKD2*, and *C21orf62* mutant cells by live cell imaging to confirm the role of these genes in mitocytosis.

## Figures and Tables

**Figure 1 cells-12-02166-f001:**
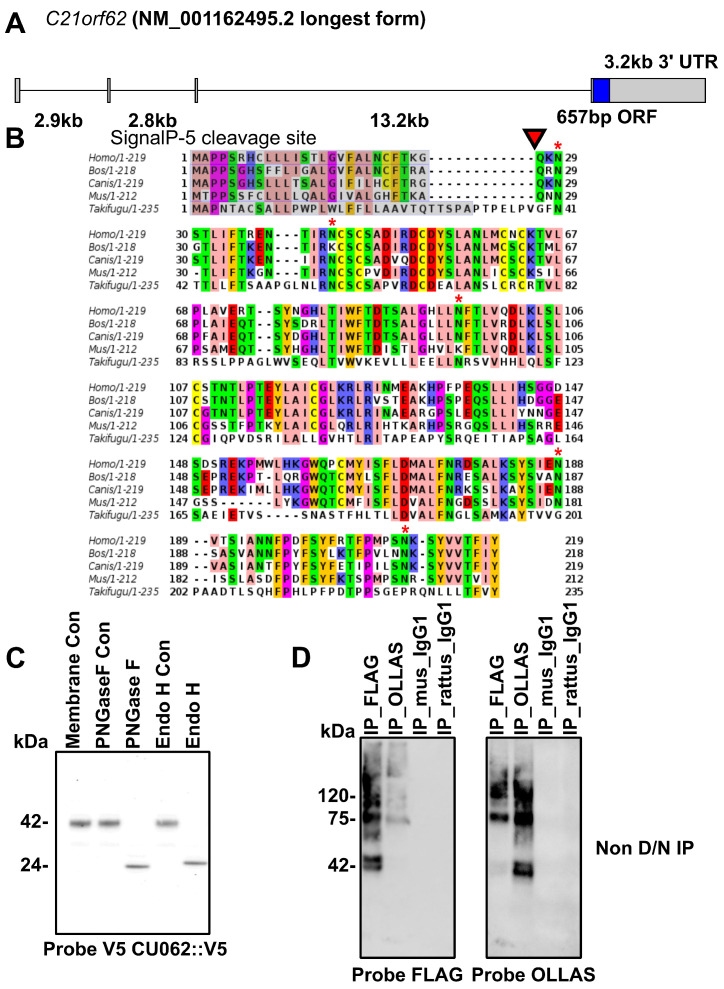
**CU062 The product of the *C21orf62* gene.** (**A**) The *C21orf62* gene on human chromosome 21 has four exons with only the last exon, exon−4, coding; (**B**) CU062 has a well-defined classical signal peptide cleavage site (red triangle). There is no transmembrane region, implying that CU062 is secreted as a soluble protein into the ER. It has five N−linked carbohydrate attachment sites (red stars). The protein was conserved from humans to fish but lacks homologs in non-vertebrates. (**C**) Transient transfection of CU062::V5 into 293t cells resulted in a highly glycosylated protein with 18kDa of N−linked carbohydrates, which are both EndoH and PNGase F sensitive; (**D**) CU062 can interact homotypically. Non-reducing SDS−PAGE analysis showed that the FLAG antibody can IP 42 kDa CU062::FLAG and dimeric CU062::OLLAS/CU062::FLAG; the reciprocal IP was also successful. Oligomers are visible up to 160 kDa.

**Figure 2 cells-12-02166-f002:**
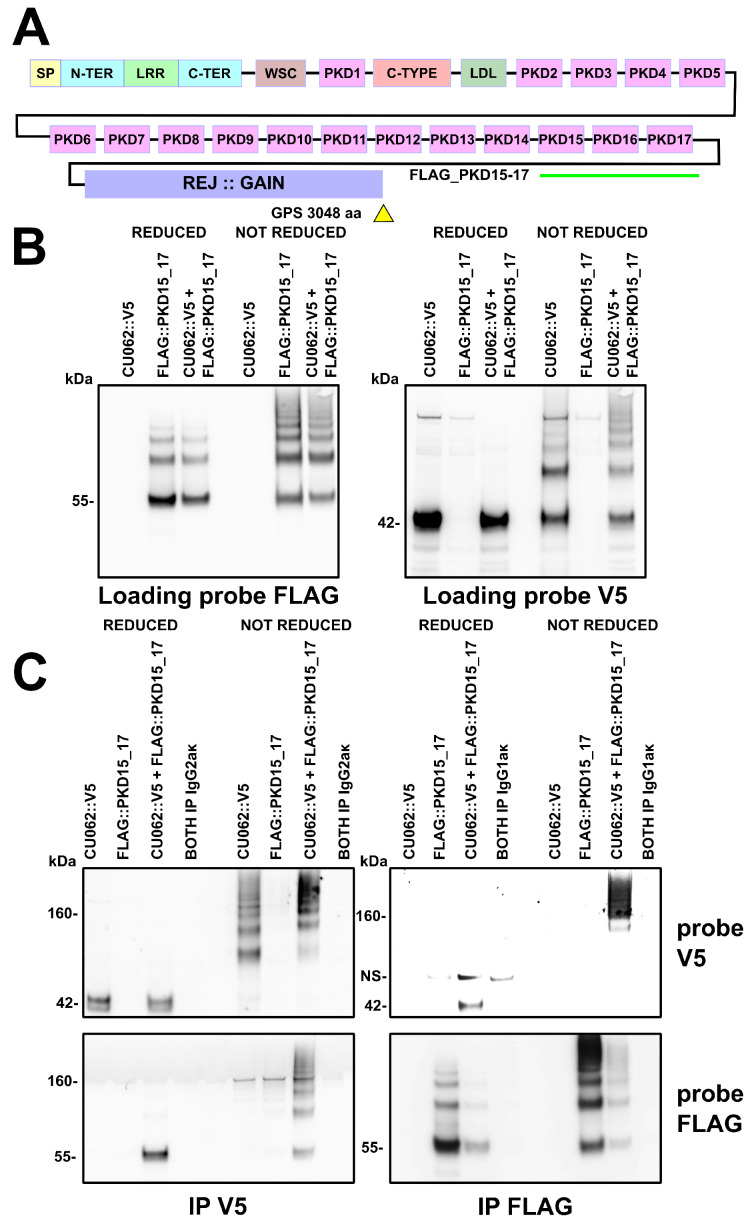
**Exploration of the interaction between CU062 and the terminal three PKD domains of PC1.** PKD domains 15, 16, and 17 have a high-affinity for CU062. (**A**) A map of the ectodomain of PC1 showing the positions of PKD repeats 15–17. (**B**) CU062::V5 and FLAG::PKD 15_17 constructs were transiently expressed in 293t cells for 10 h and resolved under reducing and non-reducing conditions. Both CU062 and the PKD array appear to homo-oligomerize as multimers of 42 kDa and 55 kDa, respectively, even after treatment with 0.5% LiDS and a temperature of 65 °C. (**C**) The PKD repeats and CU062 can reciprocally IP each other. The CU062/PKD complex appeared to be very large in the CU062::V5 IP probed for FLAG::PKD and the FLAG::PKD IP probed for CU062::V5 lanes, implying a higher-order complex (under non-reducing conditions). Myeloma IgGs of appropriate isotypes were used as controls. IgG1κ was used as a control for FLAG M2 and IgG2aκ was used as a control for SV5−Pk1.

**Figure 3 cells-12-02166-f003:**
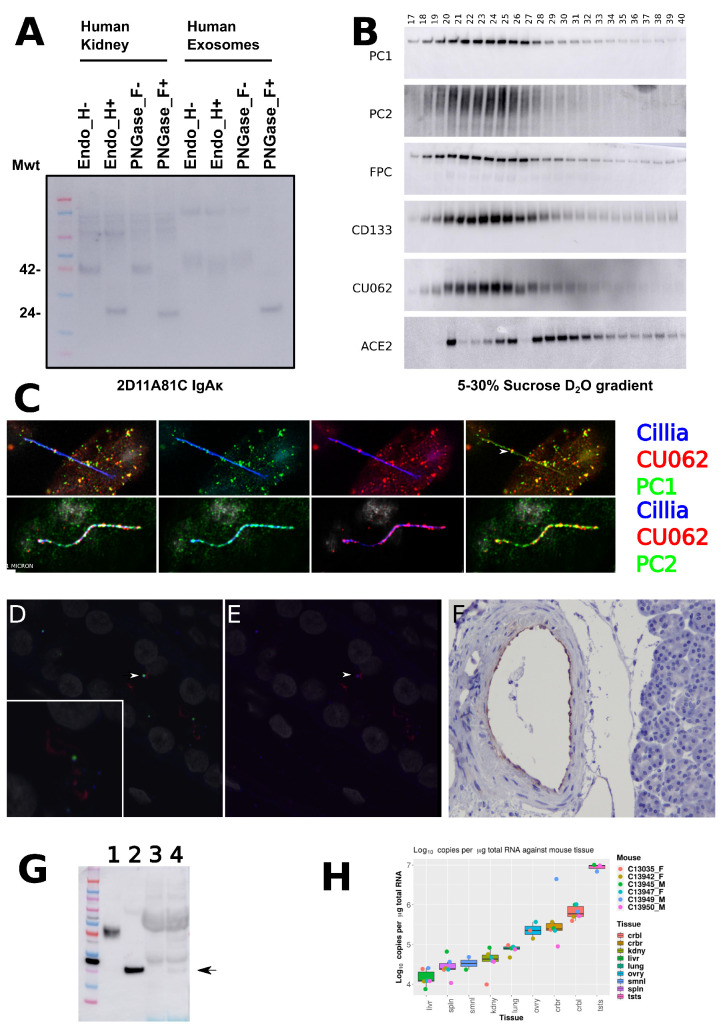
**The CU062 protein is present in the kidney and in urine exosome-like vesicles and localizes with PC1 on the primary cilia.** (**A**) Western blot of human kidney membrane proteins and urinary exosomes treated with and without EndoH and PNGase F. Kidney CU062 was sensitive to both enzymes, implying ER localization, whereas exosomal CU062 was EndoH-resistant, yet PNGase F-sensitive, implying post-Golgi processing. Monoclonal antibody (mAb) CJW_2D11 (IgAκ) was used at 2 μg m$L^{-1}$. (**B**) CU062 and PC1 co-resolve on urinary PKD−ELVs on a 5–30% sucrose D${}_{2}$O ‘float up’ gradient (ultracentrifuged at 200,000× *g* for 24 h), FPC = fibrocystin, CD133 = prominin, and ACE2 = angiotensin−converting enzyme−2 (ACE2 is not on PKD−ELVs but on another vesicle fraction). Low density fractions are on the left-hand side and high density is on the right-hand side. (**C**) Primary cilia on WT HPRE cells, acetylated α−tubulin (IgG2bκ), in blue, is used for the PC1 IF; ARL13b (polyclonal rabbit), in blue, is used for the PC2 IF. PC1 (7E12 IgG1κ), green, PC2 (CJW_9D5 IgG2bκ), green, and CU062 (CJW_2D11 IgAκ), red; see Appendix A. Note that CU062 appears to associate as a red ring around the green PC1 on 100 nm PKD−ELVs associated with the primary cilium (white arrow). See also Appendix A. (**D**) Four color confocal analyses of a normal pediatric WT human kidney, DAPI nuclei, gray, PC1 (7E12), green, primary cilia (acetylated α−tubulin), red, and CU062, blue. (**E**) The same image without PC1. Both PC1 and CU062 are present on PKD−ELVs on the primary cilia of the CD (white arrows). (**F**) On a wide-ranging adult tissue survey, CU062 was observed only in the vascular endothelium of large muscular blood vessels (a large arteriole in the human pancreas) and in the *vasa recta* in the kidney. (**G**) Western blot probed with anti-CU062 CJW_6F3 (IgG1κ). Lanes 1 and 2, exosomes from 1 mL of urine; lanes 3 and 4, exosomes from 1mL of human plasma. Lanes 1 and 3, untreated; lanes 2 and 4, treated with PNGase F. Arrow indicates deglycosylated CU062 in plasma. (**H**) Log${}_{10}$ copies of mouse CU062 mRNA per μg of total cellular RNA. Tissues, crbl = cerebellum, crbr = cerebrum, kdny = kidney, livr = liver, lung = lung, ovry = ovary, smnl = seminal vesicle, spln = spleen and tsts = testes. All RT negative reactions failed to amplify.

**Figure 4 cells-12-02166-f004:**
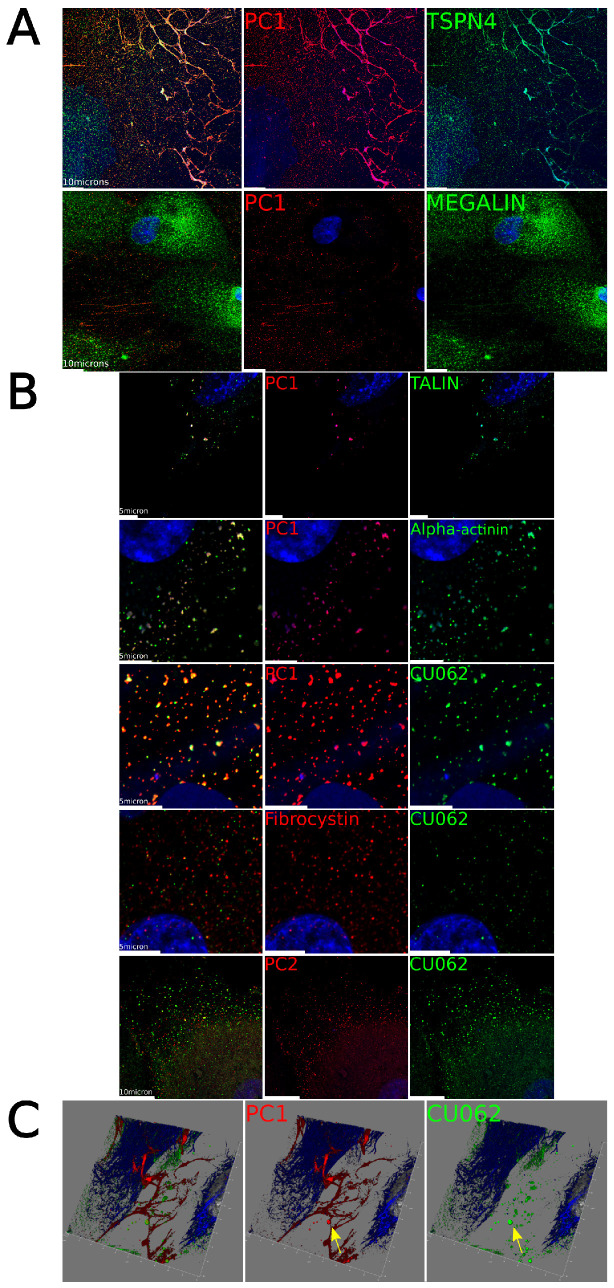
**PC1 and CU062 localize on FAs, RFs, and migrasomes in nonconfluent cells.** (**A**) Top panel, PC1, red, colocalized with tetraspanin-4, green, a marker for retraction fibers and migrasomes. The bottom panel shows that the PC1+, red, RFs and migrasomes are shed from megalin (*LRP2*) + PT cells, green. DAPI (nuclei), blue. (**B**) PC1, fibrocystin, and CU062 colocalized with the focal adhesion markers, talin and α−actinin. DAPI (nuclei), blue. Fibrocystin and PC2 only partially colocalized with CU062 but were always in close proximity to the CU062 signal (but within 100 nm). Scale bars at 10 μm. (**C**) 3D reconstruction of PC1+ retraction fibers, red, CU062 + migrasomes, green, α-acetylated tubulin, blue, and DAPI-positive nuclei, gray. CU062 coated PC1+ migrasome, yellow arrow.

**Figure 5 cells-12-02166-f005:**
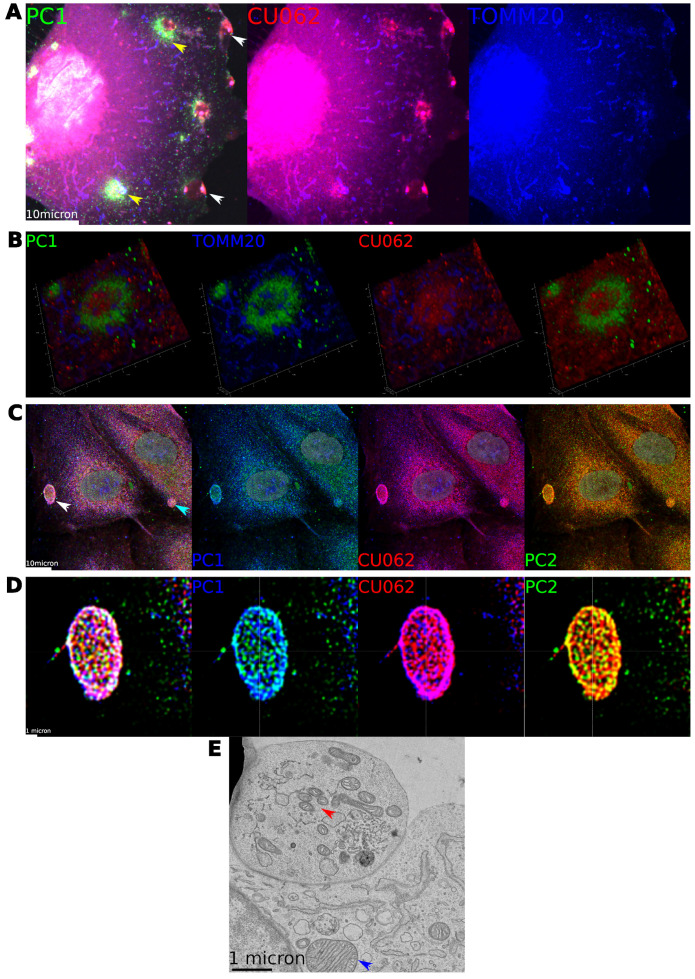
**PC1 and CU062 distribution in cells treated with 2 μM CCCP.** (**A**) Primary human PT cell treated with CCCP for 9 h: PC1 (7E12), green, CU062 (CJW_2D11), red, and TOMM20, blue. Note the ring-like structures (FRs) stained for PC1, green. These were associated with disintegrating mitochondria, blue, and an inner core of CU062, red; (yellow arrows). Note the accumulations of CU062 and the mitochondria at the edge of the cell (white arrows). (**B**) 3D reconstruction of a FR. Note the swollen mitochondria associated with the green ring of PC1; see also Appendix A. (**C**) Shedding of large extracellular vesicles in cells treated with CCCP, white and blue arrows, stained with PC1 (7E12), blue, CU062 (CJW_2D11), red, and PC2 (CJW_9D5), green. (**D**) High-resolution image with lightning deconvolution showing the reticular association of PC1, PC2, and CU062 proteins. (**E**) A TEM of an apical vesicle, 4.3μm by 3.5μm, in a primary renal epithelial cell treated with CCCP. Note the relatively normal mitochondrion in the cell body (blue arrow), and the fragmented mitochondria in the apical vesicle (red arrow).

## Data Availability

Most of the data are available in this paper or the supplement. The corresponding author, Christopher J. Ward, will supply any data or reagents mentioned in this paper upon email request. This may require a material transfer agreement (MTA) from KUMC.

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
