# Peer review of "Polycystin-1 Interacting Protein-1 (CU062) Interacts with the Ectodomain of Polycystin-1 (PC1)"

_cells, 2023, doi:10.3390/cells12172166_

Round 1

Reviewer 1 Report

In this study authors investigated the polycystin-1 (PC1)/polycistin -1 interacting protein-1 (CU062) interaction, tissue distribution and subcellular localization of CU062 and other members of the polycystin complex (PCC). They also examined the distribution of PC1 and CU062 in the presence and absence of the mitochondrial proton gradient uncoupling agent carbonyl cyanide m-chlorphenyl-hydrazine (CCCP). At the end, they postulated that the PCC is intimately involved in sorting and transporting senescent mitochondria in mitocytosis proces, and that defects in PCC dependent mitocytosis are responsible for accumulation of senescent mitochondria observed in ADPKD.

This is an extensive and valuable research and I have some comments and suggestions.

In the section Material and methods some websites are cited with detailed protocols. It is ok, but in the subsection, Statistics cited websites should be supplemented with specific tests that were used.

Some sentences were repeated in results (lines 177-179) and discussion (lines 266-268).

It would be appropriate to state limitations of the study.

As there are many different analyses, a concise conclusion with the main points of the research is needed.

Author Response

Thank you for the timely review:

All alterations are high lighted with a yellow box in the final .pdf (\colorbox{yellow} in the .tex document). For the statistics section we used only `Tukey boxplots' to show the RT-PCR quantification data, Figure_3H. We have included the appropriate reference for this  (ggplot2). We have removed the duplicated sentence and included a short conclusion.

We removed the repeated section and added a conclusion with suggestions for future studies that will address the limitations of this study.

Reviewer 2 Report

The article is very interesting in design and content as it opens new possibilities to explain some ADPKD characteristics and it begins to investigate a new protein that could have some role in ADPKD or in polycystins function. However it is a bit difficult to read, at least in some paragraphs because of the abundance of acronyms so I suggest to repeat their explanation or to substitute them in certain points or completely with the whole word: for example immunoprecipitate instead of IP in some lines among 85, 88, 90, 102, 110.

Another help for readers could be to add a section with all acronyms explained so that they are easily understandable at any point of the article without looking for the explanation in the text.

There are many minor revisions almost all linked to acronyms:

line 23: ADPKD, it would be better autosomal dominant polycystic kidney disease (ADPKD) as it is the first time that the word appears in the text (the previous citation is in the abstract); the same for PC1 in line 25, for FAs in line 44, for PKD-ELVs in line 46 and for RFs in line 148

line 25: amino acids (aa)

line 29: transmembrane (TM)

line 34: open reading frame (ORF)

line 45: MS/MS (…)

line 77: what does KG↑QKN mean?; “not” is in bold

line 78: pI explain please

line 79-80: PNGaseF/EndoH

line 85: PC-1 correct with PC1

line 101: EndoH explain

line 118: endoplasmic reticulum (ER)

line 142: RCTE and NHPTK please explain

line 143: HPRE and PT please explain

line 166: LTA and DBA please explain

line 140: the authors tested the presence of PC1 and CU062 in human kidney tubules: what kind of specimen did they use? a kidney biopsy?

line 226: glutaraldehyde is in bold

Author Response

Thank you. All changes have been made and highlighted in yellow in the final .pdf

Reviewer 3 Report

An outstanding work analyzing CU062 protein and its interaction with PC1 in human APKD the most frequent renal genetic disease.

Author Response

Thank you.

Round 2

Reviewer 1 Report

THe authors responded to all comments.